# Patterns of Health and Health Service Use in a Prospective Cohort of Aboriginal and Torres Strait Islander Children Aged 5–9 Years Living in Urban, Regional and Remote Areas of South Australia

**DOI:** 10.3390/ijerph20126172

**Published:** 2023-06-19

**Authors:** Deirdre Gartland, Arwen Nikolof, Karen Glover, Cathy Leane, Petrea Cahir, Mohajer Hameed, Stephanie J. Brown

**Affiliations:** 1Intergenerational Health, Murdoch Children’s Research Institute, Parkville, VIC 3052, Australia; deirdre.gartland@mcri.edu.au (D.G.); arwen.nikolof@mcri.edu.au (A.N.); karen.glover@sahmri.com (K.G.); petrea.cahir@mcri.edu.au (P.C.); mohajer.hameed@latrobe.edu.au (M.H.); 2Department of Paediatrics, The University of Melbourne, Parkville, VIC 3010, Australia; 3Women and Kids Theme, South Australian Health and Medical Research Institute, Adelaide, SA 5000, Australia; 4Women’s and Children’s Health Network, SA Health, North Adelaide, SA 5006, Australia; cathy.leane2@sa.gov.au

**Keywords:** indigenous child health, heath service utilisation, primary care, prospective cohort

## Abstract

Despite longstanding recognition of disparities in Aboriginal and Torres Strait Islander child health, progress to reduce disparities is slow. To improve the capacity of policy makers to target resources, there is an urgent need for epidemiological studies providing prospective data on child health outcomes. We undertook a prospective population-based study of 344 Aboriginal and Torres Strait Islander children born in South Australia. Mothers and caregivers reported on child health conditions, use of health services and the social and familial context of the children. A total of 238 children with a mean age of 6.5 years participated in wave 2 follow-up. Overall, 62.7% of the children experienced one or more physical health conditions in the 12 months prior to wave 2 follow-up, 27.3% experienced a mental health condition and 24.8% experienced a developmental condition. The 12-month period prevalence of physical, developmental and mental health conditions was similar for children living in urban, regional and remote areas. While most children had had at least one visit with a general practitioner, some children experiencing physical, developmental and mental health conditions appear to be missing out on specialist and allied health care. Greater efforts by governments and policy makers are needed to strengthen outreach, recognition, referral and follow-up.

## 1. Introduction

Aboriginal and Torres Strait Islander people living in the country now known as Australia have sustained connections to culture, identity, knowledge, place and relationships to land and community over thousands of generations [1]. The legacy and ongoing impacts of colonisation are experienced by Aboriginal and Torres Strait Islander children and families on a daily basis [2,3]. Aboriginal and Torres Strait Islander children experience a far greater burden of childhood conditions such as ear health problems, respiratory conditions and developmental conditions than non-Aboriginal and Torres Strait Islander children [4,5]. These conditions interfere with wellbeing and areas of development, such as speech and language, and impose significant hardship on families and children themselves [5,6,7]. Historical and social factors, such as dispossession of land, decimation of languages and cultural practices, racism and discrimination, and downstream factors such as insecure and inadequate housing, financial insecurity, unemployment, intergenerational trauma and the removal of children from families contribute to ongoing disparities [8,9,10,11,12,13,14,15].

The United Nations has identified that equitable early childhood policies and programs are essential for meeting the Sustainable Development Goals and for children to reach their full potential [16,17]. In the Australian health care system, general practitioners are generally the first point of contact with health care. Although visits to general practitioners are subsidised by Medicare (Australia’s universal health insurance scheme), many private practices charge above the schedule fee, resulting in variable ‘out of pocket costs’ at the point of service. All Australian states and territories also provide universal early childhood health services, although the reach and focus of these services vary by jurisdiction. During the period of our study, South Australia provided a universal home visit by a child and family health nurse within six weeks of the birth, and an enhanced service for Aboriginal and Torres Strait Islander families with children up to the age of five. In South Australia (and other state and territory jurisdictions), a limited number of Aboriginal community-controlled health services provide primary care services (and sometimes specialist services) for local Aboriginal communities. These services are free at the point of service, and often care is provided by a mix of practitioners, including Aboriginal health workers, nurses, general practitioners and visiting specialists. Aboriginal and Torres Strait Islander children are eligible for a free annual ‘child health check’ with a general practitioner. Access to government-subsided visits with allied health professionals, such as speech pathologists and psychologists, is facilitated via general practitioner referral. However, there are often ‘out of pocket’ costs for these services unless they are provided through an Aboriginal community-controlled health service. General practitioner referral is generally also required to access specialist medical practitioners, who provide services via public hospital outpatient clinics or privately.

Despite longstanding recognition of disparities in Aboriginal and Torres Strait Islander child health outcomes, progress to reduce such disparities has been slow. Research attention has largely focused on Aboriginal and Torres Strait Islander children living in remote communities and/or in states with larger populations of Aboriginal and Torres Strait Islander people [18]. Studies involving urban Aboriginal and Torres Strait Islander children remain relatively few [19,20,21,22,23]. The Study of Environment on Aboriginal Resilience and Child Health (SEARCH) conducted in New South Wales is the largest longitudinal cohort of urban Aboriginal children (aged 0–17 years) [21]. Published findings identify high developmental risk in younger children, and high numbers of mental health-related emergency department presentations and hospital admissions among Aboriginal children and adolescents living in urban areas of New South Wales [6,22]. A smaller cross-sectional study of 273 urban Aboriginal children and young people aged 0–18 years presenting to an Aboriginal Community Paediatric Service found that the most frequent diagnoses were speech and language delay, attention deficit hyperactivity disorder and school difficulty, with high numbers of referrals for audiology, ear, nose and throat (ENT) specialists and speech therapy [23]. Notably, the authors drew attention to the low number of referrals to mental health services relative to the number of mental health diagnoses. Service shortfalls including long waiting periods, poorly defined referral pathways and a lack of culturally responsive services may compound the burden of childhood ill health and contribute to poorer health outcomes across the life course [24,25,26].

To improve the capacity of policy makers to target health resources to children and families most in need of support, there is an urgent need for high-quality epidemiological studies providing prospective data on child health outcomes and the use of health services. This study draws on data collected in the Aboriginal Families Study, a prospective mother and child cohort study investigating the health and wellbeing of 344 Aboriginal children and their mothers living in urban, regional and remote areas of South Australia [27,28]. The study was designed and conducted in partnership with the Aboriginal Health Council of South Australia, the peak body representing Aboriginal community-controlled health organisations in South Australia. Two waves of data collection have been completed to date: the first when the study children were around 6 months of age and the second when the children were aged 5–9 years.

An Aboriginal Advisory Group has provided governance for the study since 2007. Members of the group include community leaders with knowledge of South Australian Aboriginal communities and expertise in policy, government and community-controlled health services. The Aboriginal Advisory Group meets six to eight times a year to set directions for the study and support the research team to work in culturally safe ways with Aboriginal families and communities. All decisions regarding the interpretation of data, dissemination of findings and subsequent follow-up of families have been made with direction from the Aboriginal Advisory Group. The inaugural Chair of the Aboriginal Advisory Group (KG) was a study investigator on the original NHMRC grant awarded in 2011, and several other Advisory Group members were associate investigators. Over time, Aboriginal study staff and Advisory Group members have increasingly taken on roles as members of the Study Investigator team. Three members of the Aboriginal Advisory Group (K.G., C.L., A.N.) are co-authors of this paper.

Drawing primarily on data collected at wave 2 follow-up when the study children were aged 5–9 years, the aims of this paper are (1) to describe the social and familial context of the study children; (2) to investigate health conditions and patterns of health service use of children living in urban, regional and remote areas of South Australia; (3) to assess differences in health conditions and patterns of health service use of children born preterm, with a low birthweight or small for gestational age compared with children not experiencing these outcomes at birth; and (4) to assess health service use of children experiencing common childhood conditions. The study’s Aboriginal Advisory Group determined that no comparisons would be made with data for non-Aboriginal Australian children. While we have chosen to reference some comparative data in the introduction and discussion, the purpose of this paper is to examine within-cohort differences in children’s patterns of health and health service use.

## 2. Materials and Methods

### 2.1. Participants

Women were eligible to participate in the study if they gave birth to an Aboriginal and/or Torres Strait Islander child in South Australia between July 2011 and June 2013, and if they were aged 14 years or older at the time this child was born. A team of Aboriginal researchers facilitated community engagement, recruitment and follow-up of families. Women were recruited via public hospitals, community-based agencies, community events and community networks of Aboriginal research team members. In the first stage of the study, mothers were invited to complete a structured questionnaire when the study children were 4–10 months of age. Wave 2 follow-up (conducted between mid-2018 and 2020) was timed to coincide with the study children commencing primary school. All women who took part in wave 1 were eligible to complete the wave 2 follow-up questionnaire. If the study child was not living with their mother at the time of wave 2 follow-up, the child’s primary caregiver (e.g., another family member, foster caregiver) was invited to complete a modified version of the wave 2 follow-up questionnaire. Informed consent was obtained from participants (mothers and other caregivers of the study children) by Aboriginal researchers.

The study was undertaken in accordance with the NHMRC guidelines for research with Aboriginal and Torres Strait Islander communities and the South Australian Aboriginal Health Research Accord. Community and policy goals have been kept in mind right from the start. The Aboriginal Advisory Group considered very carefully what the study was asking families to do, and how the study could ensure that there was reciprocity built into study processes. At an individual level, participants were offered a gift voucher in appreciation of their contribution to each wave of follow-up. At wave 2 child follow-up, the research team facilitated referrals to services for children with identified health or developmental needs, or when requested by family members or other primary caregivers. Importantly, the Aboriginal Advisory Group and members of the research team also committed to using the findings to advocate for changes to policy and services in South Australia and nationally to benefit Aboriginal and Torres Strait Islander families. In addition, community members have been kept informed about study findings through regular community newsletters and visits to communities to provide feedback and ongoing community consultation. Further details regarding community engagement, partnership arrangements, study procedures and processes used for community feedback are available elsewhere [27,28].

### 2.2. Data Collection

The wave 1 questionnaire asked about women’s health and wellbeing, infant birth outcomes (including birthweight and gestation) and women’s views and experiences of using health services during pregnancy and the year after the birth. Information was also collected on social and contextual factors (e.g., maternal age, household composition, connections to Aboriginal language, community or clan groups). The wave 2 follow-up questionnaire asked about maternal health and wellbeing, the health and wellbeing of the study child and about the use of health services in the past 12 months. The modified version of the wave 2 questionnaire completed by other primary caregivers included a brief section focusing on the living arrangements and social context of the study child, and questions asking about the health and wellbeing of the study child and their contacts with health services in the preceding 12 months.

Mothers and other primary caregivers were asked to indicate whether the study child had experienced any of the following conditions in the past 12 months: respiratory conditions (asthma, chronic cough or wheeze, constant runny nose); ear health problems (including ear ache/infection, glue ear/fluid in ears, hole in ear drum/perforated ear drum); hearing problems (one or both ears); allergy; eczema; skin infections; eye or vision problems; burn, scald or other injury; sleep problems; anxiety/worries; or weight problems (yes/no). In a separate question, they were asked, ‘Have you been told by a doctor your child has any of the following problems: autism, attention problem (e.g., ADHD), emotional or behavioural problem, speech problem (e.g., problem with speech sounds or slow to talk), developmental problem, disability or learning difficulty’ (yes/no). To assess the use of primary health care services, mothers and primary caregivers were asked whether their child had seen a local doctor, child health nurse or Aboriginal health worker in the past 12 months. In addition, they were asked whether their child had seen any of the following specialists or allied health professionals in the past 12 months: a paediatrician (doctor for children); ear, nose and throat (ENT) specialist; audiologist (to check your child’s hearing); optometrist (to check your child’s eyes); physiotherapist (about your child’s physical development); speech therapist (about speech problems such as stuttering or not being able to say sounds or words properly); psychologist/counsellor (about your child’s emotional wellbeing or behaviour). Finally, they were asked whether they had taken the study child to a hospital emergency department or outpatient clinic in the past 12 months, and whether the study child had been admitted to hospital in this time period.

Information was also collected on a range of social and contextual factors, including family members involved in parenting the study child, household composition and living arrangements, socio-demographic characteristics, and Aboriginal and Torres Strait Islander community connections of mothers and fathers of the study children. In addition, mothers were asked to report on housing stability, stressful events, and social health issues and financial difficulties. These data were not available for children who were living with other primary caregivers.

### 2.3. Analysis

All analyses were conducted using STATA version 17 [29]. The Australian Geographical Classification System was used to classify children as living in urban, regional or remote areas of South Australia at the time of wave 1 and wave 2 follow-up based on postcodes [30]. Participant characteristics at study enrolment were compared with routinely collected data for Aboriginal infants born in South Australia during the study period to assess the representativeness of the cohort in relation to key demographic and obstetric characteristics [28,31].

For aim 1, descriptive frequencies for social and contextual factors were calculated for children living in urban, regional and remote areas of South Australia, and for the sample as a whole. For aim 2, the 12-month period prevalence of common child health conditions was calculated for children living in urban, regional and remote areas and for the sample as a whole. In addition, descriptive frequencies were calculated for the use of a range of health services in the 12 months preceding wave 2 follow-up. Logistic regression was used to assess differences in the prevalence of individual health conditions and the use of health services, comparing findings for children living in urban areas of South Australia (reference category) with children living in regional areas and remote areas of South Australia. For aim 3, the 12-month period prevalence of common child health conditions was calculated for children born preterm (<37 weeks gestation), with a low birthweight (<2500 g) or small for gestational age (<10th percentile for Australian birthweight standards) [32]. Descriptive frequencies were calculated for the use of health services in the 12 months preceding wave 2 follow-up by children born preterm, low birthweight or small for gestational age. Logistic regression was used to assess differences in the prevalence of individual health conditions and the use of health services in children experiencing adverse birth outcomes compared with health conditions and health service use in children not experiencing adverse birth outcomes (reference category). No adjustments were made for potential confounders, as the focus of the study was to examine differences in the prevalence of conditions and patterns of health service use rather than understand factors that may explain any identified differences. For aim 4, descriptive frequencies were calculated for the use of a range of health services in the 12 months preceding wave 2 follow-up for children experiencing three types of common childhood conditions: ear health problems, mental health problems and developmental conditions.

## 3. Results

### 3.1. Sample

A total of 344 eligible women enrolled in the study. Mothers ranged in age from 15 to 49 years at the time of birth of the study child and had connections with more than 35 Aboriginal and Torres Strait Islander language and community groups across Australia. At enrolment, 39% of mothers and study children were living in urban areas, 36% in regional areas and 25% in remote areas of South Australia. Comparisons with South Australian routinely collected perinatal data showed that participants are largely representative in relation to maternal age, infant birthweight and gestation, but slightly over-represent women having their first child (42.2% vs. 34.3% in routinely collected data [28,31].

As shown in Figure 1, 222 mothers and 24 other primary caregivers of the study children completed the wave 2 follow-up questionnaire, providing data on 246 children (72% of the original sample). Reasons for non-participation in wave 2 follow-up were that the mother withdrew (*n* = 23, including two with children who had passed away) or the mother/primary caregiver of the study child was unable to participate/be contacted (*n* = 75, including one mother who had passed away). At wave 2 follow-up, 44.0% of the study children were living in Adelaide (the major metropolitan city in South Australia), 36.7% were living in regional areas of South Australia and 19.2% lived in areas of the state classified as remote. The average age of the study children was 6.5 years (SD = 1.0, range 5–9 years). Slightly more than half of the children participating in wave 2 follow-up were assigned male gender at birth (55.7%); 13.9% were born preterm, 13.4% had a low infant birthweight and 19.1% were born small for gestational age. Comparisons with the original cohort demonstrate that families participating in wave 2 follow-up are largely representative of the original cohort in relation to maternal age at first birth, education, place of residence at the time of recruitment and birth outcomes of the infants (Appendix A).

In this paper, data are presented for 238 children with data available on child health conditions and place of residence at wave 2 follow-up. Most of these children were living with their mother at wave 2 follow-up and one in ten were living with another primary caregiver. In most cases, this was a family member: 17 of the study children were living with grandparent/s or aunt/s, 4 were living with their father and 3 were living in formal foster care arrangements. The average age of mothers and other primary caregivers at wave 2 follow-up was 34.2 years (SD = 8.7 years). Non-maternal primary caregivers were generally older than mothers (mean = 50.5 years and 32.4 years, respectively; t(249) = −12.3, *p* < 0.001). The majority of mothers were Aboriginal and/or Torres Strait Islander women (89.1%) and most of the study children whose primary caregiver was someone other than their mother were cared for by an Aboriginal and/or Torres Strait Islander person (18/24, 75%). Three-quarters of the study children living with primary caregivers other than their mother (75%) had been living with the same primary caregiver for three or more years. In many cases, informal arrangements had been made for the study child to live with another family member. Formal arrangements through court orders or child protection were in place for just under half of the children who were not living with their mother at wave 2 follow-up (45%).

As shown in Table 1, a variety of family members were involved in parenting the study children, including mothers, fathers, grandparents, aunts and uncles and older siblings. This was a consistent pattern across urban, regional and remote areas. One in three study children lived in a household headed by a single adult (most commonly their mother). Around half lived with two adults, and one in six with three or more adults. This pattern was consistent across urban, regional and remote areas. Around 59% of the study children had regular contact (daily or weekly) with their biological father. Children living with their mothers were mostly living in public housing or private rental accommodation. Almost one in three families had moved three or more times in the past five years. Over half of mothers reported that they had experienced three or more stressful events such as the death of a family member or other social health issues such as housing problems, going to court or having to leave home because of a family argument in the past 12 months. A high proportion of mothers reported financial difficulties, including not being able to pay bills on time or going without meals. Slightly more than a third of mothers had paid employment at the time of wave 2 follow-up.

### 3.2. Child Health Conditions

Addressing aim 2, Table 2 reports the 12-month period prevalence of health conditions experienced by the study children living in urban, regional and remote areas of South Australia, and for the cohort as a whole. The most common physical health conditions were ear health conditions (including infections, glue ear, perforated ear drum), respiratory conditions (including asthma, chronic cough, constant runny nose) and allergy or eczema experienced by 27.1%, 25.0% and 22.1% of children, respectively. Overall, 62.3% of the children experienced one or more physical health conditions in the 12 months prior to wave 2 follow-up. The 12-month period prevalence of individual physical health conditions was similar across urban, regional and remote areas, with some notable exceptions. While the overall prevalence of ear health problems in urban and remote areas was similar (affecting around one in three children), ear health problems were less common among children living in regional areas (affecting around one in six children). In contrast, there was a twofold increase in the odds of allergy/eczema and skin infections reported for children living in remote areas compared to those living in urban areas. 

Slightly more than a quarter of children (27.3%) experienced mental health problems in the 12 months prior to wave 2 follow-up, and a similar proportion experienced developmental problems (24.8%). The most common mental health problem reported by mothers and caregivers was anxiety or worries, affecting almost one in five children (19.6%). The most common developmental conditions were speech and language problems and learning difficulties, experienced by 16.7% and 11.5% of the children, respectively. One in fifteen children (6.4%) had been diagnosed as having autism and a similar proportion of children (6.8%) had attention difficulties. As there was substantial overlap between these diagnoses, we combined data for the purposes of comparisons across geographic locations. The 12-month period prevalence of mental health and developmental conditions was similar for children living in urban, regional and remote areas. While the confidence intervals for all estimates suggest no statistically significant differences, it is noteworthy that the odds of mental health problems were consistently lower for children living in regional and remote areas compared with children living in urban areas.

Table 3 reports the 12-month period prevalence of health conditions experienced by children born preterm, children with a low birthweight and children born small for gestational age. The most common physical health conditions experienced by children born preterm were ear health conditions (38.7%), respiratory conditions (22.6%) and eye/vision problems (20.0%). Anxiety and sleep problems were also common among the study children born preterm, experienced by 30% and 20% of children, respectively. The most common developmental concerns for children born preterm were speech and language problems (19.4%), autism/attention difficulties (19.4%), developmental issues (12.9%) and learning difficulties (12.9%). Children born preterm had twofold higher odds of experiencing ear health conditions, autism/attention difficulties, developmental issues and weight issues, though none of these differences reached statistical significance. Similar patterns were present for children born with a low birthweight. Notably, one in three (32.1%) experienced ear health conditions, more than one in four experienced anxiety (28.6%) and one in five experienced vision/eye problems (21.4%) and weight issues (21.4%). Children with a low birthweight had markedly higher odds of experiencing issues related to their weight. They also had raised odds of experiencing speech and language problems and eye/vision problems, though neither of these differences reached statistical significance.

### 3.3. Patterns of Health Service Use

Most of the study children (80.8%) had seen a general practitioner at least once in the past 12 months, 42.4% had seen a child health nurse and 47.2% had seen an Aboriginal health worker (see Table 4a). Children living in remote areas of South Australia were markedly more likely to have seen a child health nurse or Aboriginal health worker in the past 12 months compared to children living in urban areas. Over three-quarters of children living in remote areas (76.6%) had seen a child health nurse in the past 12 months compared with only one in three children living in urban (34.9%) and regional (32.5%) areas. Similarly, the majority of children living in remote areas (83.0%) had seen an Aboriginal health worker compared with one in three in urban areas (32.4%) and two in five in regional areas (45.8%). Similar patterns of use of primary care services were apparent for the study children born preterm, with a low birthweight or small for gestational age (Table 4b), with one notable exception. Children born small for gestational age were less likely to have visited a general practitioner in the past 12 months.

Almost a third of study children (31.2%) had seen a paediatrician in the 12 months prior to wave 2 follow-up, and more than one in ten (12.7%) had seen an ENT specialist (Table 4a). Children living in remote communities had higher odds of seeing an ENT specialist (21.3% vs. 12.3% in urban areas), though this difference did not reach statistical significance. Study children born preterm or with a low birthweight were more likely to have seen an ENT specialist in the past 12 months compared to children not experiencing these birth outcomes (Table 4b). The odds of seeing an ENT specialist were also raised twofold for children born small for gestational age, but this difference did not reach statistical significance. Children born preterm, with a low birthweight or small for gestational age had higher odds of visiting a paediatrician compared with children not experiencing these outcomes, but again differences were not statistically significant.

Mothers and caregivers were also asked to indicate whether the study child had attended appointments with allied health professionals in the past 12 months. One in three children (35.0%) had seen an audiologist, 30.8% had seen an optometrist and 27.0% had seen a speech pathologist (Table 4a). A smaller proportion of children had seen a psychologist (15.6%) or physiotherapist (12.2%). Children living in remote areas had twofold higher odds of having an appointment with an audiologist compared with children living in urban areas. There was no evidence of other major differences in children’s use of allied health services in relation to their place of residence (Table 4a) or birth outcomes (Table 4b), with the exception of visits to physiotherapists. Children with a low birthweight or small for gestational age had twofold higher odds of attending a physiotherapist compared with children not experiencing these birth outcomes. However, the confidence intervals for these estimates were wide and did not reach statistical significance.

One in five children (20.8%) had visited a hospital emergency department in the 12 months preceding wave 2 follow-up, 15.8% had visited a hospital outpatient clinic and 11.5% had been admitted to the hospital (Table 4a). Children living in remote areas had twofold higher odds of attending an emergency department compared to children living in urban areas, with 95% confidence limits bordering on significance. There was no evidence of other major differences in children’s use of hospital emergency departments and outpatient or inpatient services in relation to their place of residence (Table 4a) or birth outcomes (Table 4b), with the exception of visits to the emergency department by children born small for gestational age. These children appeared to be less likely to have visited a hospital emergency department in the 12 months preceding wave 2 follow-up compared with children not born small for gestational age.

Finally, we looked specifically at health service use for children experiencing ear health conditions, mental health issues and developmental conditions (Table 5). More than half (51.6%) of children experiencing ear health conditions had seen a paediatrician and/or an ENT specialist in the past 12 months. Just under half (46.9%) had been assessed by an audiologist and around two in five (39.1%) had seen a speech pathologist. While the numbers in our sample are too small for statistical comparisons, it is noteworthy that a higher proportion of children living in remote communities had been assessed by an ENT specialist and had their hearing tested by an audiologist than in urban and regional communities. The proportion of children experiencing ear health conditions seeing a speech pathologist did not differ across urban, regional or remote communities. Just under half of the children experiencing mental health conditions (47.7%) and almost two-thirds of children experiencing developmental conditions (63.8%) had seen a paediatrician in the past 12 months. A higher proportion of children with developmental conditions living in remote communities had seen a speech pathologist, physiotherapist or psychologist compared with children living in urban areas. Only around a third of children (35.4%) experiencing mental health difficulties had seen a psychologist.

## 4. Discussion

This study is the first to examine parent/caregiver-reported common health and developmental conditions and health service use of Aboriginal and Torres Strait Islander children living in urban, regional and remote areas of South Australia. We found high 12-month period prevalence of ear health conditions, respiratory conditions, allergy/eczema, anxiety and speech or language problems in 5–9-year-old children. Overall, there were few differences in the period prevalence of parent/caregiver-reported child health or developmental conditions comparing children living in urban, regional and remote areas or comparing children born preterm, with a low birthweight or small for gestational age with children not experiencing these birth outcomes.

Globally, First Nations children in both urban and remote communities suffer ear health conditions including acute otitis media earlier, more often and with more serious complications than non-Indigenous children [33,34,35,36]. Peak periods of infection in non-Indigenous children are between the ages of 6 and 24 months, and at age 4 to 5 years when children enter pre-school [33,36]. Our findings highlight the ongoing burden of ear health conditions and associated hearing problems among 5–9-year-old Aboriginal and Torres Strait Islander children living in urban, regional and remote communities. Of note, study children living in remote areas of South Australia were more likely to have been assessed by an audiologist in the past 12 months than children living in urban or regional communities. It is unclear whether the slightly higher prevalence of hearing problems identified in children living in remote communities (OR = 2.3, 95%CI 0.8–6.3) represents a real difference or higher ascertainment of hearing difficulties related to greater access to audiology. Curiously, the period prevalence of ear health problems was lower in regional communities than in urban communities. It is plausible that this reflects an under-recognition of ear health problems rather than a real difference.

Consistent with findings from the Western Australian Aboriginal Child Health Survey, we found that parents/caregivers identified significant concerns regarding children’s anxiety symptoms, especially with regard to children living in urban areas [37]. In our study, there was a clear gradient in parent/caregiver reports of children’s anxiety symptoms, with the highest level of concern in urban areas and the lowest level of concern in remote communities. While the numbers in our study are small, and significant diversity exists in the regions of South Australia categorised as remote, it is possible that smaller local communities, with stronger connections to land, language and culture, may be protective for Aboriginal children’s mental health and wellbeing [3,11]. Despite differences in parent/caregiver reports of children’s anxiety symptoms, the proportion of children attending psychologists or counsellors was similar across urban and remote communities, highlighting potential gaps in children’s access to supportive intervention, especially in urban areas. This is borne out of the mismatch between the proportion of children identified as having problems with anxiety by their primary caregiver and the smaller proportion of children with emotional/behavioural difficulties confirmed by a health professional.

Consistent with other research, our study identified generally high levels of use of primary health care services across urban, regional and remote areas of South Australia. Children in remote areas have better access to Aboriginal health workers and child health nurses via Aboriginal community-controlled health services located in remote towns and communities. These services also facilitate access to specialist paediatric services and allied health accounting for the generally higher use of these services by children living in remote areas. Other noteworthy findings include the significantly higher proportion of children born preterm or with a low birthweight who had seen an ENT specialist in the past 12 months, and apparent under-utilisation of general practitioner services and lower use of hospital emergency departments by children born small for gestational age. It is possible that higher contact with specialist paediatric services accounts for some of this difference. However, the fact that two out of five children born small for gestational age had not seen a general practitioner in the past 12 months suggests that some of these children may be missing out on routine primary health care.

Taken together, our findings suggest that while many families with children experiencing physical, mental or developmental conditions are accessing primary care and specialist services, some children appear to be missing out on the care and support that they need to achieve optimal physical health, development and emotional wellbeing. Of note, children living in remote communities appear to have better access to allied and specialist care than children living in urban and regional areas of South Australia, although it may be that this is limited to initial assessments for the identification of developmental problems. This may reflect policy priority given to the provision of primary and specialist health care to remote communities, and long waiting lists for speech pathology and other allied health services in urban and regional communities [38,39,40]. Our data suggest that equivalent or greater efforts need to be made by state and federal governments and policy makers to support Aboriginal children’s access to primary, specialist and allied health care in both urban and regional communities.

### Strengths and Limitations

The Aboriginal Families Study is the only prospective cohort of Aboriginal and Torres Strait Islander children born in South Australia and their mothers and caregivers. An Aboriginal Advisory Group established under the auspice of the Aboriginal Health Council of South Australia in 2007 has guided the study since its inception and continues to play a major role in the design and conduct of the research and the interpretation of findings. In this paper, the Aboriginal Advisory Group determined that we would focus on internal within-cohort comparisons and longitudinal analyses leveraging data collected in both waves of data collection. In part, this was to avoid any potential to reinforce a deficit discourse regarding Aboriginal and Torres Strait Islander children’s health and developmental outcomes. The focus on establishing whether there were differences in outcomes for children living in urban, regional and remote areas was strategic, as it is often assumed that children living in remote areas experience greater health disadvantage and poorer access to health services. We are mindful of the small sample size and limited power for many of the comparisons presented. It is likely that this may have resulted in some under-identification of differences in childhood conditions or health service use associated with place of residence or adverse birth outcomes. While the findings must be interpreted with caution, the study remains important as one of the few ‘community-led’, co-designed studies to provide evidence about Aboriginal and Torres Strait Islander children’s health based on parent/caregiver reports and facilitated by an Aboriginal researcher/interviewer. The study focuses on parent/caregiver perceptions of the study children’s health and wellbeing, and services that parents/caregivers have used to support their child’s health and development. In this sense, it provides a unique window into Aboriginal and Torres Strait Islander families’ experience of their children’s health in middle childhood. The lives of families in the study reflect the social and historical context of Aboriginal and Torres Strait Islander people and the legacy and ongoing impacts of colonisation. Many families were experiencing housing instability and financial and other stresses; issues likely to limit families’ access and engagement with services that are also well documented in other cohorts of Aboriginal and Torres Strait Islander children [9,41].

While it is unclear how generalisable the findings are to Aboriginal and Torres Strait Islander children living in other Australian states and territories, it is likely that many of the issues identified by our study are salient to the experiences of Aboriginal and Torres Strait Islander families living in other urban, regional and remote communities in Australia. A major weakness of our study is that it does not provide any evidence regarding the extent to which families accessing primary, specialist or allied health care received culturally appropriate support matched to their needs, nor do the findings enable us to identify optimal ways to reduce the prevalence of common childhood conditions among Aboriginal and Torres Strait Islander children or improve their access to appropriate primary, specialist and allied health care. However, the findings do point to the need to strengthen outreach, recognition, referral and follow-up of Aboriginal and Torres Strait Islander children in this age group. The research team has made the results available to the South Australian government and is participating in policy processes to enhance continuity of care protocols for Aboriginal children and families in South Australia [25].

## 5. Conclusions

Our findings highlight a high prevalence of physical, developmental and mental health conditions among Aboriginal and Torres Strait Islander children aged 5–9 years living in urban, regional and remote communities in South Australia. While many families with children experiencing physical, mental or developmental conditions are accessing primary care and specialist services, some children appear to be missing out on the care and support that they need to achieve optimal physical health, development and emotional wellbeing. Greater efforts by state and federal governments and policy makers are needed to strengthen outreach, recognition, referral and follow-up of Aboriginal and Torres Strait Islander children experiencing physical, developmental and mental health conditions, particularly in urban and regional communities. Equitable service provision matched to need requires the provision of culturally safe, accessible and affordable primary care, allied health and specialist services for Aboriginal and Torres Strait Islander children and families living in urban, regional and remote communities. Workforce innovation is also required to address workforce shortages in primary and allied health care, and to support career pathways for Aboriginal and Torres Strait Islander people in primary, specialist and allied health care.

## Figures and Tables

**Figure 1 ijerph-20-06172-f001:**
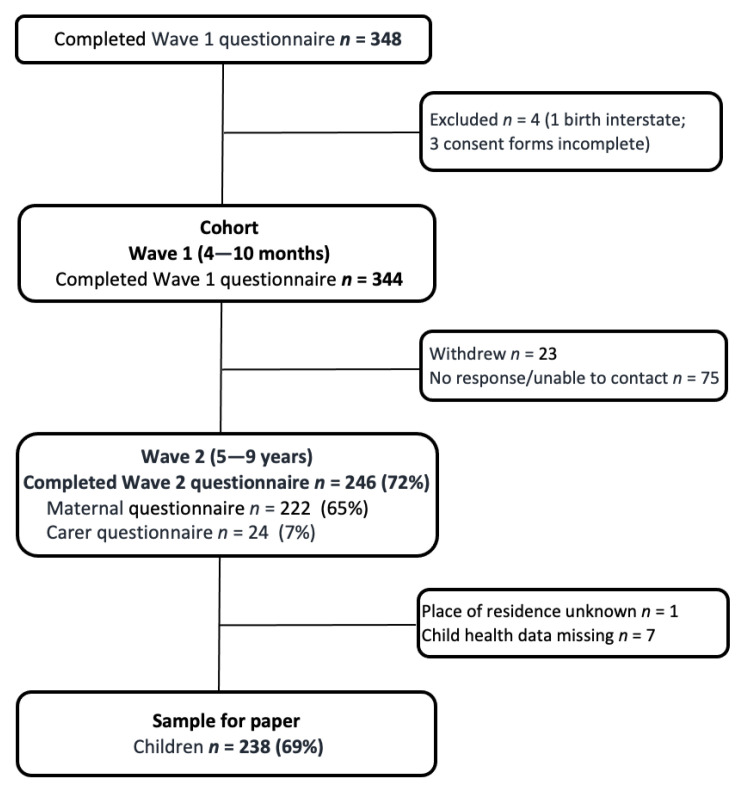
Flowchart of participation in the Aboriginal Families Study.

**Table 1 ijerph-20-06172-t001:** Social and family life of study children aged 5–9 years living in urban, regional and remote areas of South Australia (*n* = 238).

	Urban	Regional	Remote	Total
	*n* ^1^ (%)	*n* (%)	*n* (%)	*n* (%)
Family members involved in parenting study child				
Mother	103 (98.1)	83 (100)	46 (97.9)	232 (98.7)
Father	66 (62.9)	50 (60.2)	29 (61.7)	145 (61.7)
Aunts/uncles	68 (64.8)	34 (41)	27 (57.4)	129 (54.9)
Grandparents	43 (41.0)	29 (34.9)	20 (42.6)	92 (39.1)
Older siblings	37 (35.2)	22 (26.5)	18 (38.3)	77 (32.8)
Mother’s partner	32 (30.5)	27 (32.5)	16 (34)	75 (31.9)
Study child lives with				
Mother	98 (92.5)	73 (85.9)	43 (91.5)	214 (89.9)
Father (not living with mother)	1 (0.9)	3 (3.5)	0 (0)	4 (1.7)
Other primary caregiver	7 (6.6)	9 (10.6)	4 (8.5)	20 (8.4)
Study child age at wave 2 follow-up				
5–6 years	62 (58.5)	52 (61.2)	19 (40.4)	133 (55.9)
7–9 years	44 (41.5)	33 (38.8)	28 (59.6)	105 (44.1)
Indigenous status of parents				
Mother Aboriginal and/or Torres Strait Islander	96 (90.6)	72 (84.7)	44 (93.6)	212 (89.1)
Father Aboriginal and/or Torres Strait Islander	77 (72.6)	62 (74.7)	42 (89.4)	181 (76.7)
Study child spends time with biological father				
Daily	52 (49.1)	48 (60.8)	27 (60.0)	127 (55.2)
Weekly	5 (4.7)	3 (3.8)	0 (0)	8 (3.5)
Monthly	13 (12.3)	2 (2.5)	3 (6.7)	18 (7.8)
Yearly	9 (8.5)	4 (5.1)	7 (15.6)	20 (8.7)
Not at all	27 (25.5)	22 (27.8)	8 (17.8)	57 (24.8)
Number of adults in household				
One adult	36 (34.6)	32 (38.1)	10 (21.7)	78 (33.3)
Two adults	53 (51)	41 (48.8)	21 (45.7)	115 (49.1)
Three or more adults	15 (14.4)	11 (13.1)	15 (32.6)	41 (17.5)
Siblings living in household				
None	6 (5.7)	10 (11.8)	1 (2.1)	17 (7.1)
One or two	50 (47.2)	28 (32.9)	15 (31.9)	93 (39.1)
Three or four	37 (34.9)	41 (48.2)	22 (46.8)	100 (42)
Five or more	13 (12.3)	6 (7.1)	9 (19.1)	28 (11.8)
Other children in household				
None	85 (80.2)	66 (77.6)	37 (78.7)	188 (79)
One or two	14 (13.2)	14 (16.5)	7 (14.9)	35 (14.7)
Three or more	7 (6.6)	5 (5.9)	3 (6.4)	15 (6.3)
Housing tenure ^2^				
Public housing	40 (40.8)	34 (47.2)	31 (72.1)	105 (49.3)
Private rental	35 (35.7)	25 (34.7)	7 (16.3)	67 (31.5)
Own house/mortgage	20 (20.4)	12 (16.7)	4 (9.3)	36 (16.9)
Refuge	3 (3.1)	1 (1.4)	1 (2.3)	5 (2.3)
Housing stability (past 5 years) ^2^				
Lived in same house	21 (21.6)	13 (18.3)	14 (32.6)	48 (22.7)
Moved once or twice	39 (40.2)	38 (53.5)	20 (46.5)	97 (46)
Moved three or more times	37 (38.1)	20 (28.2)	9 (20.9)	66 (31.3)
Stressful events and social health issues (past 12 months) ^2^				
None	13 (13.3)	7 (9.6)	5 (11.6)	25 (11.7)
One or two	25 (25.5)	24 (32.9)	16 (37.2)	65 (30.4)
Three or more	60 (61.2)	42 (57.5)	22 (51.2)	124 (57.9)
Financial stress (past 12 months) ^2^				
Could not pay bills on time	55 (56.1)	35 (48.6)	14 (32.6)	104 (48.8)
Went without meals	13 (13.3)	12 (16.9)	6 (14.3)	31 (14.7)
Unable to heat/cool home	12 (12.2)	11 (15.5)	6 (14)	29 (13.7)
Pawned/sold something	24 (24.5)	11 (15.5)	5 (11.6)	40 (18.9)
Assistance from welfare organisation	21 (21.4)	14 (19.7)	3 (7)	38 (17.9)
Mothers’ age at wave 2 follow-up ^2^				
20–24 years	5 (4.7)	6 (7.1)	1 (2.1)	12 (5)
25–29 years	43 (40.6)	23 (27.1)	10 (21.3)	76 (31.9)
30–34 years	30 (28.3)	23 (27.1)	19 (40.4)	72 (30.3)
≥35 years	28 (26.4)	33 (38.8)	17 (36.2)	78 (32.8)
Mothers’ highest level education ^2^				
University Degree	11 (11.5)	4 (5.6)	0 (0)	15 (7.1)
Diploma/Certificate	28 (29.2)	15 (20.8)	13 (31)	56 (26.7)
Completed Year 12	14 (14.6)	14 (19.4)	4 (9.5)	32 (15.2)
Year 10 or less	43 (44.8)	39 (54.2)	25 (59.5)	107 (51)
Mothers’ participation in paid employment ^2^				
Working full time	16 (16.3)	10 (13.7)	7 (16.3)	33 (15.4)
Working part time	21 (21.4)	12 (16.4)	12 (27.9)	45 (21.1)
Not working in paid employment	60 (61.2)	51 (69.9)	24 (55.8)	135 (63.4)
Study child at birth				
Infant birthweight				
≥2500 g	88 (85.4)	68 (87.2)	42 (93.3)	198 (87.6)
<2500 g	15 (14.6)	10 (12.8)	3 (6.7)	28 (12.4)
Infant gestation				
≥37 weeks (not preterm)	85 (86.7)	68 (85)	40 (87)	193 (86.2)
<37 weeks (preterm)	13 (13.3)	12 (15)	6 (13)	31 (13.8)
Infant birthweight for gestational age				
≥10th percentile	77 (83.7)	60 (81.1)	34 (79.1)	171 (81.8)
<10th percentile	15 (16.3)	14 (18.9)	9 (20.9)	38 (18.2)
Total	106 (100)	85 (100)	47 (100)	238 (100)

^1^ Numbers may not add up to total due to missing data. ^2^ Data only available from mothers (not 24 other primary caregivers).

**Table 2 ijerph-20-06172-t002:** Health conditions experienced by Aboriginal children living in urban, regional and remote areas of South Australia (n = 238).

	Cohort (*n* = 238)	Urban (*n* = 106)		Regional (*n* = 85)		Remote (*n* = 47)	
	*n* (%)	*n* (%)	OR [95%CI]	*n* (%)	OR [95%CI]	*n* (%)	OR [95%CI]
Physical Health							
Respiratory condition ^1^	59 (25.0)	29 (27.4)	1.0 [ref]	20 (23.8)	0.8 [0.4–1.6]	10 (21.7)	0.7 [0.3–1.7]
Allergy/eczema	52 (22.1)	19 (17.9)	1.0 [ref]	18 (21.7)	1.3 [0.6–2.6]	15 (32.6)	2.2 [1.0–4.9]
Skin infection	28 (11.9)	12 (11.3)	1.0 [ref]	5 (6.0)	0.5 [0.2–1.5]	11 (23.9)	2.5 [1.0–6.1]
Injury/burn/scald	23 (9.8)	8 (7.5)	1.0 [ref]	10 (12.0)	1.7 [0.6–4.5]	5 (10.9)	1.5 [0.5–4.8]
Ears health condition ^2^	64 (27.1)	34 (32.1)	1.0 [ref]	14 (16.7)	0.4 [0.2–0.9]	16 (34.8)	1.1 [0.5–2.3]
Hearing problem	24 (10.2)	9 (8.5)	1.0 [ref]	7 (8.3)	1.0 [0.3–2.8]	8 (17.4)	2.3 [0.8–6.3]
Eye/vision issue	31 (13.2)	17 (16.0)	1.0 [ref]	12 (14.5)	0.9 [0.4–2.0]	2 (4.3)	0.2 [0.1–1.1]
Weight issue (underweight/overweight)	16 (6.8)	7 (6.6)	1.0 [ref]	5 (6.0)	0.9 [0.3–3.0]	4 (8.7)	1.3 [0.4–4.8]
Any physical health issue	147 (62.3)	67 (63.2)	1.0 [ref]	49 (58.3)	0.8 [0.5–1.5]	31 (67.4)	1.2 [0.6–2.5]
Mental Health							
Emotional/behavioural issue ^3^	21 (9.0)	12 (11.3)	1.0 [ref]	6 (7.4)	0.6 [0.2–1.7]	3 (6.4)	0.5 [0.1–2.0]
Anxiety/worries	46 (19.6)	25 (23.6)	1.0 [ref]	15 (18.1)	0.7 [0.3–1.5]	6 (13.0)	0.5 [0.2–1.3]
Sleep issue	32 (13.6)	15 (14.2)	1.0 [ref]	11 (13.3)	0.9 [0.4–2.1]	6 (13.0)	0.9 [0.3–2.5]
Any mental health issue	65 (27.3)	34 (32.1)	1.0 [ref]	20 (23.5)	0.7 [0.3–1.2]	11 (23.4)	0.6 [0.3–1.4]
Development							
Autism/attention problem ^3^	23 (9.8)	8 (7.5)	1.0 [ref]	12 (14.8)	2.1 [0.8–5.5]	3 (6.4)	0.8 [0.2–3.3]
Developmental issue ^3^	15 (6.4)	8 (7.5)	1.0 [ref]	4 (4.9)	0.6 [0.2–2.2]	3 (6.4)	0.8 [0.2–3.3]
Speech or language problem ^3^	39 (16.7)	17 (16.0)	1.0 [ref]	15 (18.5)	1.2 [0.6–2.6]	7 (14.9)	0.9 [0.4–2.4]
Learning difficulty ^3^	27 (11.5)	14 (13.2)	1.0 [ref]	10 (12.3)	0.9 [0.4–2.2]	3 (6.4)	0.4 [0.1–1.6]
Disability ^3^	9 (3.8)	2 (1.9)	1.0 [ref]	4 (4.9)	2.7 [0.5–15.1]	3 (6.4)	3.5 [0.6–22.0]
Any developmental issue	58 (24.8)	26 (24.5)	1.0 [ref]	24 (29.6)	1.3 [0.7–2.5]	8 (17.0)	0.6 [0.3–1.5]

^1^ Includes asthma, chronic cough, constant runny nose. ^2^ Includes earache/infection, glue ear/fluid in ears, hole in ear drum/perforated ear drum. ^3^ Confirmed by health practitioner; other conditions reported by caregiver.

**Table 3 ijerph-20-06172-t003:** Health conditions experienced by study children born preterm, low birthweight and/or small for gestational age (n = 238).

	Preterm (<37 Weeks Gestation) *n* = 31/224		Low Birthweight (<2500 g) *n* = 28/226		Small for Gestational Age (<10th percentile) ^4^ *n* = 39/209	
	*n* (%)	OR [95%CI]	*n* (%)	OR [95%CI]	*n* (%)	OR [95%CI]
Physical Health						
Respiratory condition ^1^	7 (22.6)	0.9 [0.4–2.2]	7 (25.0)	1.0 [0.4–2.6]	4 (10.8)	0.3 [0.1–1.0]
Allergy/eczema	3 (10.0)	0.4 [0.1–1.2]	1 (3.6)	0.1 [0.0–0.8]	3 (8.1)	0.3 [0.1–0.9]
Skin infection	1 (3.3)	0.2 [0.0–1.6]	1 (3.6)	0.2 [0.0–1.8]	2 (5.4)	0.3 [0.1–1.4]
Injury/burn/scald	3 (10.0)	1.0 [0.3–3.6]	1 (3.6)	0.3 [0.0–2.5]	2 (5.4)	0.5 [0.1–2.2]
Ear health condition ^2^	12 (38.7)	1.9 [0.9–4.3]	9 (32.1)	1.4 [0.6–3.3]	6 (16.2)	0.5 [0.2–1.4]
Hearing problem	4 (12.9)	1.5 [0.5–4.8]	3 (10.7)	1.1 [0.3–4.1]	3 (8.1)	0.8 [0.2–3.1]
Eye/vision issue	6 (20.0)	1.7 [0.6–4.5]	6 (21.4)	1.9 [0.7–5.0]	6 (16.2)	1.2 [0.5–3.3]
Weight issue (underweight/overweight)	4 (13.3)	2.8 [0.8–9.5]	6 (21.4)	5.0 [1.7–15.2]	4 (10.8)	2.4 [0.7–8.6]
Any physical health issue	17 (54.8)	0.7 [0.3–1.5]	16 (57.1)	0.8 [0.4–1.8]	18 (48.6)	0.5 [0.2–1.1]
Mental Health						
Emotional/behavioural difficulty ^3^	3 (9.7)	1.1 [0.3–3.9]	2 (7.1)	0.7 [0.2–3.2]	5 (13.2)	1.5 [0.5–4.5]
Anxiety/worries	9 (30.0)	1.8 [0.8–4.4]	8 (28.6)	1.7 [0.7–4.0]	6 (16.2)	0.7 [0.3–1.8]
Sleep issue	6 (20.0)	1.7 [0.6–4.7]	5 (17.9)	1.4 [0.5–4.0]	4 (10.8)	0.8 [0.3–2.5]
Any mental health issue	12 (38.7)	1.9 [0.8–4.1]	12 (42.9)	2.1 [0.9–4.7]	10 (26.3)	0.9 [0.4–2.1]
Development						
Autism/attention problem ^3^	6 (19.4)	2.6 [0.9–7.3]	4 (14.3)	1.6 [0.5–5.3]	5 (13.2)	1.7 [0.6–4.9]
Developmental issue ^3^	4 (12.9)	2.7 [0.8–9.1]	3 (10.7)	1.8 [0.5–6.9]	3 (7.9)	1.5 [0.4–5.9]
Speech or language problem ^3^	6 (19.4)	1.3 [0.5–3.4]	7 (25.0)	2.0 [0.8–5.1]	10 (26.3)	2.4 [1.0–5.5]
Learning difficulty ^3^	4 (12.9)	1.1 [0.4–3.5]	4 (14.3)	1.2 [0.4–3.9]	6 (15.8)	1.6 [0.6–4.2]
Disability ^3^	2 (6.5)	1.8 [0.4–9.1]	2 (7.1)	2.1 [0.4–10.5]	2 (5.3)	1.5 [0.3–7.7]
Any developmental issue	8 (25.8)	1.1 [0.4–2.5]	9 (32.1)	1.6 [0.7–3.7]	13 (34.2)	1.8 [0.9–3.9]

^1^ Includes asthma, chronic cough, constant runny nose. ^2^ Includes earache/infection, glue ear/fluid in ears, hole in ear drum/perforated ear drum. ^3^ Confirmed by health practitioner, other conditions reported by caregiver. ^4^ Excludes twins.

**Table 4 ijerph-20-06172-t004:** Health service utilisation by (a) family location and (b) children born preterm, of low birthweight or small for gestational age.

a. Health service utilisation in the past 12 months by study children living in urban, regional and remote areas of South Australia (*n* = 238)							
	Cohort	Urban		Regional		Remote	
	*n* (%)	*n* (%)	OR [95%CI]	*n* (%)	OR [95%CI]	*n* (%)	OR [95%CI]
Primary Health Care							
General practitioner	189 (80.8)	87 (82.1)	1.0 [ref]	63 (75.9)	0.7 [0.3–1.4]	39 (86.7)	1.4 [0.5–3.8]
Child health nurse	100 (42.4)	37 (34.9)	1.0 [ref]	27 (32.5)	0.9 [0.5–1.7]	36 (76.6)	6.1 [2.8–13.4]
Aboriginal health worker	111 (47.2)	34 (32.4)	1.0 [ref]	38 (45.8)	1.8 [1.0–3.2]	39 (83.0)	10.2 [4.3–24.1]
Specialist Medical Practitioner							
Paediatrician	74 (31.2)	32 (30.2)	1.0 [ref]	23 (27.4)	0.9 [0.5–1.6]	19 (40.4)	1.6 [0.8–3.2]
Ear nose and throat specialist	30 (12.7)	13 (12.3)	1.0 [ref]	7 (8.3)	0.7 [0.2–1.7]	10 (21.3)	1.9 [0.8–4.8]
Allied Health Practitioner							
Audiologist	83 (35.0)	34 (32.1)	1.0 [ref]	25 (29.8)	0.9 [0.5–1.7]	24 (51.1)	2.2 [1.1–4.5]
Speech pathologist	64 (27.0)	23 (21.7)	1.0 [ref]	28 (33.3)	1.8 [0.9–3.4]	13 (27.7)	1.4 [0.6–3.0]
Optometrist	73 (30.8)	31 (29.2)	1.0 [ref]	30 (35.7)	1.3 [0.7–2.5]	12 (25.5)	0.8 [0.4–1.8]
Physiotherapist	29 (12.2)	12 (11.3)	1.0 [ref]	9 (10.7)	0.9 [0.4–2.3]	8 (17.0)	1.6 [0.6–4.2]
Psychologist/counsellor	37 (15.6)	18 (17.0)	1.0 [ref]	10 (11.9)	0.7 [0.3–1.5]	9 (19.1)	1.2 [0.5–2.8]
Hospital Care							
Child attended emergency department	49 (20.8)	17 (16.0)	1.0 [ref]	19 (22.9)	1.6 [0.7–3.2]	13 (27.7)	2.0 [0.9–4.6]
Child admitted hospital	27 (11.5)	10 (9.5)	1.0 [ref]	10 (11.9)	1.3 [0.5–3.2]	7 (15.2)	1.7 [0.6–4.8]
Child attended hospital outpatients	37 (15.8)	17 (16.3)	1.0 [ref]	13 (15.5)	0.9 [0.4–2.1]	7 (15.2)	0.9 [0.4–2.4]
b. Health service utilisation by Aboriginal children born preterm, low birthweight and/or small for gestational age) (*n* = 238)							
		Preterm (<37 Weeks Gestation) *n* = 31/224		Low Birthweight (<2500 g) *n* = 28/226		Small for Gestational Age (<10th percentile) ^1^ *n* = 39/209	
		*n* (%)	OR [95%CI]	*n* (%)	OR [95%CI]	*n* (%)	OR [95%CI]
Primary Health Care							
General practitioner		24 (77.4)	0.8 [0.3–2.0]	21 (75.0)	0.7 [0.3–1.7]	23 (62.2)	0.3 [0.1–0.6]
Child health nurse		14 (45.2)	1.2 [0.5–2.5]	12 (42.9)	1.1 [0.5–2.4]	12 (31.6)	0.6 [0.3–1.4]
Aboriginal health worker		19 (61.3)	1.9 [0.9–4.1]	11 (39.3)	0.7 [0.3–1.5]	13 (34.2)	0.5 [0.2–1.0]
Specialist Medical Practitioner							
Paediatrician		12 (38.7)	1.5 [0.7–3.2]	13 (46.4)	2.1 [0.9–4.6]	14 (36.8)	1.4 [0.7–3.0]
Ear nose and throat specialist		9 (29.0)	4.2 [1.7–10.6]	7 (25.0)	3.1 [1.2–8.3]	6 (15.8)	2.1 [0.7–5.8]
Allied Health Practitioner							
Audiologist		12 (38.7)	1.2 [0.6–2.6]	10 (35.7)	1.1 [0.5–2.5]	12 (31.6)	0.9 [0.4–1.9]
Speech pathologist		11 (35.5)	1.7 [0.7–3.7]	8 (28.6)	1.1 [0.5–2.8]	11 (28.9)	1.2 [0.6–2.7]
Optometrist		11 (35.5)	1.2 [0.5–2.7]	10 (35.7)	1.4 [0.6–3.1]	12 (31.6)	1.0 [0.5–2.2]
Physiotherapist		5 (16.1)	1.5 [0.5–4.3]	6 (21.4)	2.3 [0.8–6.3]	7 (18.4)	2.2 [0.8–5.7]
Psychologist/counsellor		7 (22.6)	1.6 [0.6–4.2]	6 (21.4)	1.6 [0.6–4.2]	5 (13.2)	0.9 [0.3–2.5]
Hospital Care							
Child attended emergency department		5 (16.1)	0.7 [0.2–1.8]	3 (10.7)	0.4 [0.1–1.4]	3 (7.9)	0.3 [0.1–0.9]
Child admitted to hospital		4 (12.9)	1.2 [0.4–3.7]	1 (3.6)	0.3 [0.0–2.0]	1 (2.6)	0.2 [0.0–1.3]
Child attended hospital outpatients		7 (23.3)	1.8 [0.7–4.5]	4 (14.8)	0.9 [0.3–2.7]	4 (10.8)	0.6 [0.2–1.8]

^1^ Excludes twins.

**Table 5 ijerph-20-06172-t005:** Health service utilisation by study children aged 5–9 years experiencing common childhood conditions (*n* = 238).

	Urban	Regional	Remote	Total
	*n* (%)	*n* (%)	*n* (%)	*n* (%)
Ear health condition (*n* = 64)				
General practitioner	28 (82.4)	11 (78.6)	15 (93.8)	54 (84.4)
Child health nurse	12 (35.3)	7 (53.8)	12 (75.0)	31 (49.2)
Aboriginal health worker	18 (52.9)	9 (69.2)	14 (87.5)	41 (65.1)
Paediatrician	13 (38.2)	6 (42.9)	9 (56.3)	28 (43.8)
Ear nose and throat (ENT) specialist	10 (29.4)	5 (35.7)	7 (43.8)	22 (34.4)
Paediatrician and/or ENT specialist	16 (47.1)	8 (57.1)	9 (56.3)	33 (51.6)
Audiologist	13 (38.2)	7 (50.0)	10 (62.5)	30 (46.9)
Speech pathologist	13 (38.2)	6 (42.9)	6 (37.5)	25 (39.1)
Hospital outpatient department	7 (21.2)	3 (21.4)	4 (25.0)	14 (22.2)
Hospital emergency department	5 (14.7)	7 (53.8)	4 (25.0)	16 (25.4)
Mental health issue (*n* = 65)				
General practitioner	29 (85.3)	15 (75.0)	10 (90.9)	54 (83.1)
Child health nurse	11 (32.4)	6 (30.0)	8 (72.7)	25 (38.5)
Aboriginal health worker	13 (39.4)	11 (55.0)	9 (81.8)	33 (51.6)
Paediatrician	14 (41.2)	11 (55.0)	6 (54.5)	31 (47.7)
Psychologist/counsellor	12 (35.3)	7 (35.0)	4 (36.4)	23 (35.4)
Hospital outpatient department	10 (30.3)	6 (30.0)	2 (20.0)	18 (28.6)
Hospital emergency department	5 (14.7)	4 (20.0)	4 (36.4)	13 (20.0)
Developmental issue (*n* = 58)				
General practitioner	20 (76.9)	21 (87.5)	7 (87.5)	48 (82.8)
Child health nurse	10 (38.5)	6 (25.0)	8 (100.0)	24 (41.4)
Aboriginal health worker	12 (46.2)	11 (45.8)	8 (100.0)	31 (53.4)
Paediatrician	18 (69.2)	13 (54.2)	6 (75.0)	37 (63.8)
Audiologist	10 (38.5)	8 (33.3)	5 (62.5)	23 (39.7)
Speech pathologist	14 (53.8)	18 (75.0)	5 (62.5)	37 (63.8)
Physiotherapist	8 (30.8)	4 (16.7)	3 (37.5)	15 (25.9)
Psychologist/counsellor	10 (38.5)	5 (20.8)	4 (50.0)	19 (32.8)
Hospital outpatient department	9 (34.6)	7 (29.2)	1 (14.3)	17 (29.8)
Hospital emergency department	3 (11.5)	7 (29.2)	2 (25.0)	12 (20.7)

## Data Availability

Data cannot be shared publicly per the agreement between study investigators and the Aboriginal Health Council of South Australia to maximise participant privacy and confidentiality and protect Indigenous data sovereignty. Data sharing is subject to approval by the study’s Aboriginal Advisory Group and Investigator team. Applications will be considered in the context of papers in progress, compliance with conditions of ethics approval and consent, and potential benefits to Indigenous communities. Interested researchers are invited to submit a request via the Aboriginal Families Study (afs@mcri.edu.au) or to contact the Principal Investigator, Professor Stephanie Brown (stephanie.brown@mcri.edu.au).

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
