# Peer review of "Patterns of Health and Health Service Use in a Prospective Cohort of Aboriginal and Torres Strait Islander Children Aged 5–9 Years Living in Urban, Regional and Remote Areas of South Australia"

_ijerph, 2023, doi:10.3390/ijerph20126172_

Round 1
Reviewer 1 Report
Thank you for the opportunity to review this article.
It is an important prospective study that set out to follow a large cohort (n=238) of First Nations children in South Australia from across urban to remote settings. The decision to take a prospective study is excellent as it enabled the authors to seek specific data about health issues of concern.
The academic English is of a very high standard and the article is extremely polished; well done to the authors.
However, I do strongly recommend a range of changes, outlined below. Without these changes, I do not feel that the paper should be published.
These recommended changes are:
- Abstract: please revise the wording to include comparative data to the general population, and also to strengthen the focus of the recommendation for ‘greater efforts’ to be made (who holds responsibility? What are the specific actions required?).
- Authors: please clarify whether any authors are first Nations Peoples and/or whether cultural guidance was provided by senior staff for the project design and the write up of this article.
- Methods: please describe what the participants gained for their involvement in the study (to address the requirement of beneficence when undertaking research involving First Nations’ Peoples).
- Results: similar to the abstract, please contexualise the findings by comparing these to the general child population of Australia. The current data has no ability to be compared and thus to know whether the rates of diseases etc are similar or different.
- Framing: there is an active discussion in Indigenous health research regarding deficit and strengths-based communication framing. This paper in its current form is very much in a deficit frame. When First Nations Peoples read this article, would they consider that their community/ Peoples are positively and respectfully portrayed?
- Wider context: there are many explanations of why the results are high form many of the indicators measured. This includes both current and historical issues that still impact lives today- from ongoing aspects of colonisation and racism. These are key determinants of health for First Nations peoples in Australia, and this should ideally been explained as context for the results.
- Conclusion: similar to the Abstract, it is insufficient to state that ‘greater efforts’ need to be made to change the current situation. Please be bolder and more specific: who holds responsibility? what are the specific actions required?
I look forward to reading the revised paper, and thank the authors for their contribution.
Reviewer 2 Report
This is a timely, well written and very interesting paper. Given this is a community led co-designed research project I'm interested in understanding who the Indigenous authors are on the paper. The results are interesting even though statistical power is limited and certainly useful for key stakeholders, researchers and policy makers.
Reviewer 3 Report
This is a very interesting and well-done paper focused on patterns of health and health service use in a prospective cohort of Aboriginal and Torres Strait Islander children aged 5-9 years living in urban, regional, and remote areas of South Australia. I only have a few comments.
Introduction. a) Very excellent background. B) Objective was described in lines 72 to 74. Then, in lines 85 to 90 again. They need to be combined or, please differentiate the meaning.
Material and methods. a) It has a strong study design. b) How is access to health services in Australia? Is the primary care service free? And the specialized care? c) Authors refer they estimated prevalence (line 165, line 169, line 172, line 177), but if the study design is a cohort, it should be incidence. Please explain.
Results. Table 5 should have a statistical test (chi-square).
Discussion. Ok
